# The Distinct Effects of the Mitochondria-Targeted STAT3 Inhibitors Mitocur-1 and Mitocur-3 on Mast Cell and Mitochondrial Functions

**DOI:** 10.3390/ijms24021471

**Published:** 2023-01-12

**Authors:** Anastasia N. Pavlyuchenkova, Maria A. Chelombitko, Artem V. Fedorov, Maria K. Kuznetsova, Roman A. Zinovkin, Ehud Razin

**Affiliations:** 1Faculty of Bioengineering and Bioinformatics, Lomonosov Moscow State University, 119234 Moscow, Russia; 2Belozersky Institute of Physico-Chemical Biology, Lomonosov Moscow State University, 119992 Moscow, Russia; 3The “Russian Clinical Research Center for Gerontology”, Ministry of Healthcare of the Russian Federation, Pirogov Russian National Research Medical University, 129226 Moscow, Russia; 4Department of Cell Biology and Histology, Biology Faculty, Lomonosov Moscow State University, 119234 Moscow, Russia; 5Department of Biochemistry and Molecular Biology, The Faculty of Medicine, Hebrew University of Jerusalem, Jerusalem 91120, Israel

**Keywords:** mast cell, RBL-2H3 cells, mitochondria, mitochondria-targeted curcuminoids, Mitocur-1, Mitocur-3

## Abstract

There is accumulating evidence that mitochondria and mitochondrial STAT3 are involved in the activation of mast cells. The mitochondria-targeted curcuminoids Mitocur-1 and Mitocur-3 have been suggested to reduce antigen-dependent mast cell activation by inhibiting mitochondrial STAT3. The aim of the current work was to investigate the mechanisms of action of these mitocurcuminoids on mast cells and mitochondrial functions. The pretreatment of rat basophilic leukemia cells RBL-2H3 with Mitocur-1 and Mitocur-3 decreased antigen-dependent degranulation but did not affect spontaneous degranulation. Both compounds caused mitochondrial fragmentation and increased mitochondrial ROS. Inhibition of Drp1 prevented mitochondrial fragmentation induced by Mitocur-3 but not by Mitocur-1. The antioxidant N-acetylcysteine inhibited mitochondrial fission induced by Mitocur-1 but not Mitocur-3. Mitochondrial fragmentation caused by Mitocur-3 but not Mitocur-1 was accompanied by activation of Drp1 and AMPK. These data suggest a distinct mechanism of action of mitocurcuminoids on the mitochondria of RBL-2H3 cells: Mitocur-3 stimulated AMPK and caused Drp1-dependent mitochondrial fragmentation, while Mitocur-1-induced mitochondrial fission was ROS-dependent. This difference may contribute to the higher toxicity of Mitocur-3 compared to Mitocur-1. The findings contribute to further drug development for inflammatory and allergic diseases.

## 1. Introduction

Mast cells (MCs) represent an important cell population of the connective tissue that maintains its homeostasis [1]. MCs also participate in the regulation of innate and adaptive immune responses [2]. The role of MCs in the pathogenesis of various inflammatory and allergic diseases is well established. Allergies are usually managed by avoiding allergens and using treatments such as epinephrine, antihistamines, anti-inflammatory medications (corticosteroids), decongestants, leukotriene inhibitors, the anti-IgE mAb omalizumab, and allergy shots (immunotherapy). New therapeutic approaches include Toll-like receptor agonists, cytokine blockers, specific cytokine receptor antagonists and transcription factor modulators targeting Syk kinase, peroxisome proliferator-activated receptor gamma (PPARγ), and nuclear factor kappa B (NF-kB) [3]. Despite a seemingly vast repertoire of anti-allergic remedies, there is an urgent need for the development of new approaches for the modulation of MC functions.

There is a growing body of evidence that mitochondria are involved in the activation of MCs [4]. MC degranulation necessitates mitochondrial translocation to the site of exocytosis [5]. This fact assumes that some mitochondrial functions are needed for MC exocytosis. Different inhibitors of mitochondrial electron transport chain (ETC) and uncouplers of oxidative phosphorylation (OXPHOS) and respiration are able to decrease MC activation [6]. Inhibition of mitochondrial pyruvate dehydrogenase leads to the inhibition of mitochondrial respiration and subsequently decreases MC degranulation [7].

The transcription factor signal transducer and activator of transcription 3 (STAT3) has been shown to be localized in mitochondria of MCs and to play an important role in MC activation [8]. A natural polyphenol resveratrol inhibits MC activation by inhibiting the phosphorylation of mitochondrial and nuclear Erk1/2 and STAT3 [9]. STAT3 is an important transcription factor that is activated in response to various cytokines and growth factors. High constitutive activation of STAT3 is typical for some types of cancer, and contributes to oncogenesis in part by preventing apoptosis [10,11].

Mitochondrial STAT3 can increase OXPHOS activity, and also may be involved in reducing reactive oxygen species (ROS) production, preventing mPTP opening, increasing Ca^2+^ levels, and inhibiting mitophagy [12,13,14,15]. The selective STAT3 inhibitor Stattic and the mitochondria-targeted inhibitors Mitocur-1 and Mitocur-3 (curcumin conjugated with triphenylphosphonium lipophilic cations) suppress both degranulation and cytokine production by MCs [7,16]. Mitocur-1 and Mitocur-3 have previously been shown to be more toxic to breast cancer cells compared to non-transformed mammary epithelial cells. Treatment of tumor cells with mitochondria-targeted curcuminoids (mitocurcuminoids) leads to significant ROS generation, a drop in mitochondrial membrane potential (ΔΨm), inhibition of Akt and STAT3 phosphorylation and an increase in Erk1/2 phosphorylation [17]. Curcumin by itself affects the function of mitochondria due to ROS generation [18]. The energy metabolism of tumor cells can be regulated by curcumin likely through the inhibition of ATP synthase activity, which leads to an increase in ΔΨm and induction of AMP-activated protein kinase (AMPK) phosphorylation due to the altered ATM/AMP ratio [19]. 

It is known that AMPK plays a central role in the regulation of cellular metabolism. AMPK is rapidly activated in response to mitochondrial stress and involved in mitochondrial fission [20]. It was shown that AMPK pathway negatively regulates MC activation and anaphylaxis [21,22]. Activation of AMPK before antigen-dependent stimulation blocks the interaction of Lyn and Syk with the FcεRI β and FcεRI γ subunits of the FcεRI receptor, resulting in inhibition of FcεRI-dependent signaling [22].

Based on the above data, we hypothesized that Mitocur-1 and Mitocur-3 can reduce the antigen-dependent activation of MCs by modulation of mitochondrial function and AMPK activity. The aim of the current research was to study the effects of these curcuminoids on MC signaling, mitochondrial function and morphology. 

## 2. Results

The results of cell survival studies indicate that incubation of RBL-2H3 cells for 3 h with Mitocur-1 at concentrations above 6 μM led to a decrease in cell viability (Figure 1). Mitocur-3 reduced cell survival starting at a concentration of 3 μM (Figure 1). According to these data, we have chosen the non-toxic concentration of 1 μM for both compounds for further experiments. 

Pretreatment of RBL-2H3 cells with 1 μM of Mitocur-1 and Mitocur-3 for 3 h led to a 1.5-fold decrease in antigen-dependent degranulation and did not affect spontaneous degranulation (Figure 2). 

The pretreatment of cells with Mitocur-1 and Mitocur-3 did not affect Ser 727 phosphorylation of STAT3 in unstimulated RBL-2H3 cells and reduced phosphorylation in activated MCs (Figure 3). These results fully support previous findings that these compounds prevent antigen-dependent MC activation by inhibiting mitochondrial STAT3 [16].

Next, we investigated the effect of substances on the mitochondria of RBL-2H3 cells. Visualization of mitochondria using the in vivo fluorescent dyes MitoTracker Green showed that the pretreatment of cells with Mitocur-1 and Mitocur-3 for 3 h caused mitochondrial fragmentation and swelling (Figure 4). Mitochondrial fission is mediated by activation of GTPase Drp1 [23]. Moreover, Drp1 activation is accompanied by an elevation of total and mitochondrial ROS (mtROS) levels [24,25]. Pretreatment of cells with the Drp1 inhibitor mDivi-1 at concentration of 25 μM for 1 h prevented mitochondrial fragmentation induced by Mitocur-3 but not Mitocur-1 (Figure 4). Preincubation of cells with the antioxidant N-acetylcysteine (NAC) at concentration of 5 mM for 1 h inhibited mitochondrial fission induced by Mitocur-1 but not Mitocur-3 (Figure 4). 

Mitochondrial fragmentation is associated with decreased ΔΨm and elevated mtROS level [6,26]. To study the effect of mitocurcuminoids on ΔΨm and mtROS in RBL-2H3 cells, we have used fluorescent dyes TMRM, and MitoTracker Orange CM-H_2_TMRos, respectively. Unexpectedly, incubation of RBL-2H3 cells with Mitocur-3 resulted in a dramatic increase in TMRM fluorescence intensity (Figure 5), indicating an elevation in ΔΨm. Treatment of cells with Mitocur-1 also led to an increase in the ΔΨm, though not statistically significant (Figure 5). The detection of MitoTracker Orange CM-H_2_TMRos by flow cytometry fluorescence revealed that both compounds increased mtROS levels (Figure 6). 

We have additionally measured mtROS level in RBL-2H3 cells by using MitoCLox. MitoCLox is a ratiometric mitochondria-targeted fluorescence probe for the detection of cardiolipin peroxidation in mitochondria, which is a reliable marker for mitochondrial oxidative stress [27]. Treatment of cells with Mitocur-1 and Mitocur-3 resulted in an increase in the level of cardiolipin peroxidation (Figure 7).

Mitochondrial fragmentation is known to be mediated by GTPase Drp1, which is phosphorylated at serine 616 and dephosphorylated at serine 637 [28]. We measured the level of Drp1 serine 637 phosphorylation by Western blot analysis. Treatment of RBL-2H3 cells with Mitocur-3 led to a decrease in the level of Drp1 637 serine phosphorylation (Figure 8), which indicated activation of Drp1. Mitocur-1 had no effect on the activity of Drp1 (Figure 8). These results are consistent with the results obtained using mDivi-1 inhibitor (Figure 4).

As mentioned above, AMPK is rapidly activated in response to mitochondrial stress and involved in mitochondrial fission [20]. We estimated AMPK kinase activity by measuring the level of AMPK phosphorylation at threonine 172. Mitocur-3 but not Mitocur-1 activated AMPK (Figure 8). Since excessive mitochondrial dysfunction can lead to apoptosis [29], we measured caspase 3 cleavage by Western-blotting. Mitocurcuminoids did not increase the level of cleaved caspase 3 in RBL-2H3 cells (Figure 8).

Fragmentation of mitochondria may precede mitophagy [30]. To study the effect of mitocurcuminoids on the number of mitochondria in RBL-2H3 cells, we measured the relative content of mitochondrial DNA using qPCR. The relative content of mitochondrial DNA in the cells slightly decreased after incubation with Mitocur-3; however, this change was not significant (Figure 9). Mitocur-1 had no effect on these parameters (Figure 9). These results indicate that mitocurcuminoids do not reduce the number of mitochondria.

## 3. Discussion

Recently, we have shown that the mitocurcuminoids Mitocur-1 and Mitocur-3 suppress antigen-dependent MC activation by inhibiting mitochondrial STAT3 [16]. However, the exact mechanism of their action remained unexplored. In our recent study, we have demonstrated that both mitocurcuminoids caused mitochondrial fragmentation and oxidative stress, but their mechanisms of action appear to be different. 

Thus, Mitocur-3 caused Drp1-dependent fragmentation of mitochondria (Figure 4), which is accompanied by an increase in the ΔΨm (Figure 5) and the production of mtROS (Figure 6 and Figure 7). The increase in ΔΨm is often caused by enhanced production of proton gradient by functional ETC and a decreased activity of ATP synthase, which uses these protons for energy production. It is likely that Mitocur-3 affects cellular energy metabolism due to ATP synthase inhibition. This suggestion is indirectly confirmed by the fact that curcumin inhibits ATP-synthase activity in tumor cells [19]. Mitocur-3 also induced activation of AMPK kinase (Figure 8), which is known to mediate mitochondrial fission in response to energy stress [20]. At the same time, there is evidence that AMPK activation inhibits activation of MCs [21,22,31]. Since this kinase is a mitochondrial stress sensor [20], mitochondrial dysfunction caused by Mitocur-3 can act as an AMPK activator and inhibit antigen-dependent MC activation due to that. Besides, AMPK can induce mitophagy [32] and mitochondrial biogenesis [33]. Mitochondrial stresses may activate mitochondria-associated AMPK, resulting in mitophagy activation [34]. Nevertheless, the relative content of mitochondrial DNA did not change in the cells after incubation with Mitocur-3 (Figure 9).

Mitocur-1 induced ROS-dependent mitochondrial fragmentation, but had no effect on AMPK activation, ΔΨm change, or amount of mitochondria. Mitocur-1 has a similar chemical backbone structure to Mitocur-3 but contains two methoxy-groups. Probably, this subtle difference is responsible for the distinct effects of these compounds.

Our previous research and existing literature data suggest that antigen-dependent MC degranulation is accompanied by mitochondrial fragmentation [5,26]. We have shown that mitocurcuminoids cause mitochondrial fragmentation, do not affect spontaneous cell degranulation, and inhibit antigen-dependent degranulation (Figure 2). There is also evidence that uncouplers of OXPHOS and respiration can inhibit MC degranulation while simultaneously inducing mitochondrial fission [6]. Based on this, mitochondrial fragmentation itself probably does not trigger degranulation and does not enhance antigen-dependent degranulation of MCs. Thus, inducers of mitochondrial fragmentation may cause such a strong dysfunction of mitochondria that they become unable to perform their functions during antigen-dependent activation of MCs. Moreover, mitochondria are apparently not the main source of energy for MC degranulation. Thus, our previous data suggest that mitochondrial fragmentation during the degranulation of RBL-2H3 cells is characterized by a decrease in ΔΨm and mitochondrial ATP [26]. These observations are consistent with the observed increase in glycolysis processes during the antigen-dependent degranulation of MCs obtained from mice bone marrow [35]. Similar metabolic reprogramming was also shown for effector T cells—their differentiation is accompanied by mitochondrial fragmentation and a decrease in the activity of the mitochondrial electron transport chain, which leads to a decrease in the activity of OXPHOS and stimulation of aerobic glycolysis processes [36]. 

A possible role of mitochondria during MC degranulation may be related to calcium entry regulation, which is critical for antigen-dependent degranulation. For instance, the antibacterial and antifungal agent triclosan has an uncoupling activity. Triclosan treatment of unstimulated RBL-2H3 cells results in lowered ΔΨm and ATP production, increased ROS levels, and mitochondrial fragmentation. Triclosan inhibits microtubule polymerization and mitochondrial translocation to the plasma membrane in the stimulated MCs, as well as the entry of extracellular Ca^2+^. Probably, triclosan prevents antigen-dependent MC degranulation by inhibiting mitochondrial translocation, which may prevent the activation of calcium release-activated channels, and subsequent Ca^2+^ entry [37,38]. Based on this, we can assume that mitochondrial dysfunction induced by mitocurcuminoids can impair calcium entry. This hypothesis requires further experimental confirmation.

The effects of both mitocurcuminoids on antigen-dependent degranulation of RBL-2H3 cells are similar [16]. However, according to our data, Mitocur-3 induces more pronounced mitochondrial dysfunction Compared to Mitocur-1. Probably, it indicates a limited role of mitochondria in MC degranulation. At the same time, the functional activity of mitochondria may play a more critical role for cell survival at a later stage of antigen-dependent activation, namely at the stage of antigen-dependent cytokine production. Thus, our previous data indicate a stronger inhibitory effect of Mitocur-3 on antigen-induced TNF cytokine production compared to Mitocur-1 [16]. Within the framework of this work, we have demonstrated that Mitocur-3 is more toxic to RBL-2H3 cells than Mitocur-1 (Figure 1), which is most likely due to the fact that severe mitochondrial dysfunction can lead to mitochondrial-mediated death [39].

The following limitation applies to this study: since we used RBL-2H3 cells as a model of MCs, our knowledge is limited to in vitro data.

## 4. Materials and Methods

### 4.1. Cell Culture

All experiments were performed with rat basophilic leukemia cell line RBL-2H3 (purchased from ATCC). Cells were cultured in the medium containing 70% alpha modification of Minimum Essential Medium Eagle (α-MEM, PanEco, Moscow, Russia) and 20% RPMI-1640 (PanEco, Moscow, Russia) supplemented with 10% fetal bovine serum (FBS, Thermo Fisher Scientific, Waltham, MA, USA) and 2 mM L-glutamine (PanEco, Moscow, Russia) at 37 °C with 5% CO_2_. 

### 4.2. Cytotoxicity Assay

Cytotoxic activity of Mitocur-1 and Mitocur-3 was estimated by the resazurin assay. RBL-2H3 cells were plated in a 96-well plate (10,000 cells/well) and treated with 1–6 µM of mitocurcuminoids for 3 h at 37 °C and 5% CO_2_. After washing in cell medium for 24 h, 25 µL of resazurin was added to 250 µM, and the cells were incubated for 3 h at 37 °C and 5% CO_2_. The fluorescence of resorufin (the reduced form of resazurin) in the samples was determined with a Fluoroskan Ascent microplate fluorometer (Thermo Scientific, Waltham, MA, USA) using 560/590 nm (excitation/emission) filter settings.

### 4.3. Beta-Hexosaminidase Release

RBL-2H3 cells were plated on 24-well plates (100,000 cells/well) and treated with 1 μM of Mitocur-1 and Mitocur-3 for 3 h at 37 °C and 5% CO_2_. β-hexosaminidase activity in RBL-2H3 cells was used as a typical marker of MC degranulation and measured by registration of the fluorescent signal produced by the cleavage of N-acetylglucosamine residues from 4-methylumbelliferyl-N-acetyl- β-D-glucosaminide [40]. Protocol was modified as previously described [26].

### 4.4. Mitochondria Visualization with MitoTracker Green 

Cells were plated on 35 mm confocal dishes (200,000 cells/dish) and treated with 1 μM of Mitocur-1 and Mitocur-3 for 3 h at 37 °C and 5% CO_2_. After treatment, mitochondria visualization with MitoTracker Green (Invitrogen, Waltham, MA, USA) was performed as previously described [26].

### 4.5. Mitochondrial Membrane Potential Assay

Cells were plated in a 6-well plate (200,000 cells/well) and treated with 1 μM of Mitocur-1 and Mitocur-3 for 3 h at 37 °C and 5% CO_2_. After treatment with mitocurcuminoids, cells were incubated with 50 nM TMRM (Thermo Scientific, Waltham, MA, USA) for 30 min to detect mitochondrial membrane potential. Fluorescence in the PE channel was then detected by flow cytometry. 

### 4.6. Reactive Oxygen Species Detection Assay

The effect of Mitocur-1 and Mitocur-3 on ROS production of RBL-2H3 cells was measured using MitoTracker Orange CM-H_2_TMRos (Thermo Scientific, Waltham, MA, USA) and MitoCLox, a ratiometric fluorescent probe reporting cardiolipin peroxidation in living cells [27]. Cells were plated in a 6-well plate (200,000 cells/well) and treated with 1 μM of Mitocur-1 and Mitocur-3 for 3 h at 37 °C and 5% CO_2_. After treatment with mitocurcuminoids, cells were incubated with 500 nM MitoTracker Orange CM-H_2_TMRos for 30 min to detect ROS and fluorescence in the PE channel was then detected by flow cytometry. In 12 h of washing with the cell medium after treatment with mitocurcuminoids, cells were stained with MitoCLox for 30 min. MitoCLox fluorescence in the FITC and PE channels was then detected by flow cytometry. 

### 4.7. Flow Cytometry

The flow cytometric measurements were performed on the Amnis^®^ FlowSight^®^ Imaging Flow Cytometer (Luminex, Austin, TX, USA) kindly provided by the Moscow State University Development Program PNR5. The Amnis^®^ FlowSight^®^ Imaging Flow Cytometer was equipped with a 488 nm laser (60 mW) and a SSC laser (10 mW). Cells were removed from the culture vessel surface into suspension with trypsin/versene (1:1) 5 min at 3000× *g*, +4 °C and resuspended in 50 µL of PBS. For all probes, the integrated intensity of each cell was measured. Flow cytometric data analysis was performed with the IDEAS software ver. 6.3 (Luminex, Austin, TX, USA).

### 4.8. Western Immunoblotting

Cell lysis, protein separation and transfer onto polyvinylidene difluoride (PVDF) membranes were performed as previously described [26]. The membranes were incubated with the following primary monoclonal antibodies: anti-STAT3, anti-STAT3 (Ser727) (Cell Application, San-Diego, CA, USA), anti-AMPK, anti-phospho-AMPK (Thr172), anti-caspase 3, anti-DRP1 (Cell Signaling Technology, Danvers, MA, USA), and anti-phospho-DRP1 (Ser637) (Thermo Scientific, Waltham, MA, USA), followed by the incubation with HRP-conjugated goat anti-mouse IgG and anti-rabbit IgG (Sigma, St. Louis, MI, USA) secondary antibodies. HRP activity was developed by ChemiDocTM XR+ System (BioRad, Hercules, CA, USA) using a luminol-based enhanced chemiluminescence (ECL) HRP substrate Super-Signal West Dura (Thermo Scientific, Waltham, MA, USA). Densitometric analysis was performed using image processing Image Lab software version 5.2.1.

### 4.9. DNA Isolation and Quantitative PCR (qPCR)

Total DNA was isolated from the cultured RBL-2H3 cells using Quick-DNA Miniprep Kit (Zymo Research, Irvine, CA, USA) according to the manufacturer’s instructions. DNA was amplified using specific primers and the PowerUpTM SYBRTM Green Master Mix (Life Technologies, Carlsbad, CA, USA, A25778), according to the manufacturer’s instructions.

Primer sequences (5′-3′) were as follows:

rMito3969 F—ACCACCAATATGAGTAGGA;

rMito3969 R—TTGTAGTTGAGGATGATGAC; 

rLINE F—AGACATACAAGAAGCCTAC;

rLINE R—TCTGGTGTGATTCTGATAG.

The reaction was performed in a C1000 Touch PCR thermal cycler (BioRad, Hercules, CA, USA) using the following protocol: 95 °C 3′ → (95 °C 15″ → 56 °C 15″ → 72 °C 15″) × 45 cycles. Nuclear DNA and mitochondrial DNA (mtDNA) were amplified at efficiencies of 1.03 and 1.20, respectively. The mtDNA/nuclear DNA ratio was calculated using the 2^−ΔΔCt^ method with calculated PCR efficiencies.

### 4.10. Statistical Analysis

Statistical analysis was performed using one-way analysis of variance (ANOVA) with Dunnett’s test. Graphs were plotted using the GraphPad Prism 6 software, and data are presented as the mean ± SD.

## 5. Conclusions

The results of the present study suggest that the mitocurcuminoids Mitocur-1 and Mitocur-3, which are thought to inhibit antigen-dependent MC degranulation through downregulation of mitochondrial STAT3, act on the mitochondria of RBL-2H3 cells prior to antigen stimulation. Mitocur-1 and Mitocur-3 cause mitochondrial fragmentation and oxidative stress, but their mechanisms of action differ. Thus, mitochondrial fission induced by Mitocur-1 is ROS dependent but Drp1 independent, while Mitocur-3 causes AMPK activation and Drp1-dependent mitochondrial fragmentation. The findings help to improve our understanding of the mechanisms of action of mitochondrial-derived curcuminoids on MCs and their mitochondrial functions. This information can be used to develop new drugs for the treatment of inflammatory and allergic diseases. 

## Figures and Tables

**Figure 1 ijms-24-01471-f001:**
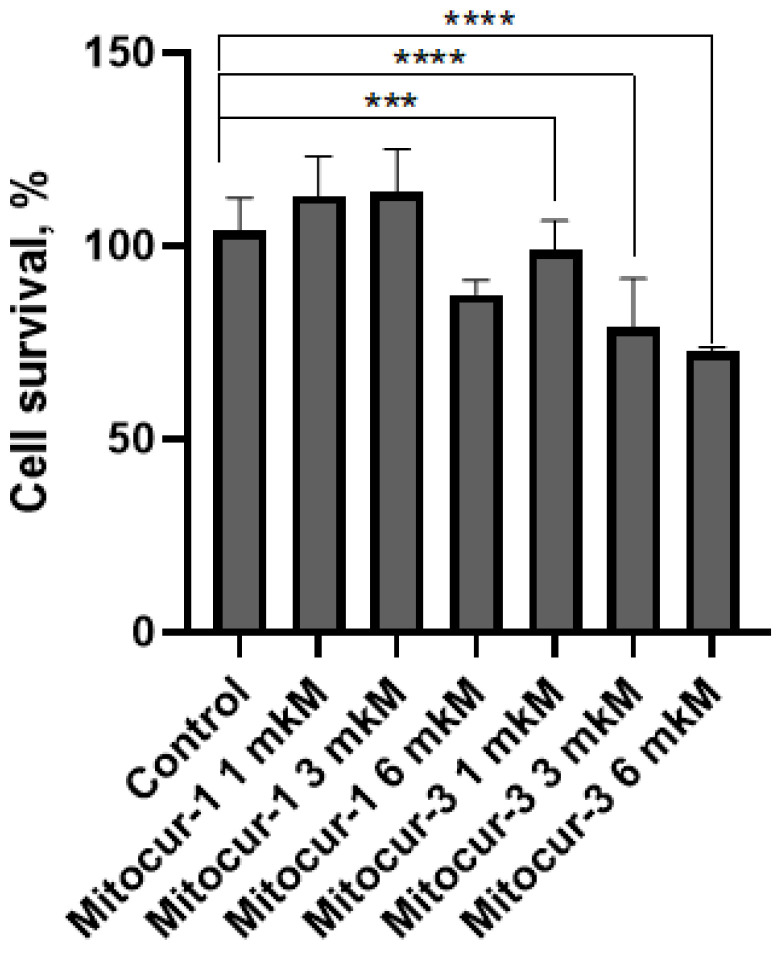
Effect of different concentrations of mitochondria-targeted curcuminoids Mitocur-1 and Mitocur-3 on cell survival. The cells were incubated with 1–6 µM of mitocurcuminoids for 3 h. After washing in cell medium for 24 h, viability was measured by resazurin. The results are presented as the mean ± SD (*n* = 4). *** *p* ≤ 0.001, and **** *p* ≤ 0.0001 as compared to untreated cells calculated by one-way ANOVA, Dunnett’s test.

**Figure 2 ijms-24-01471-f002:**
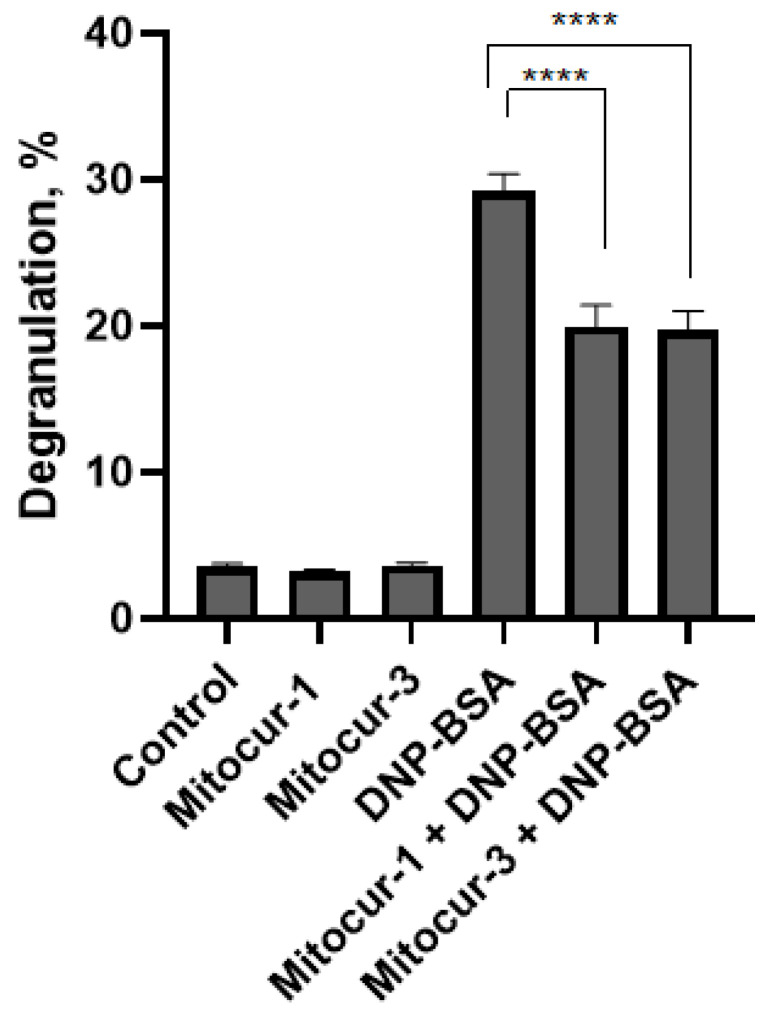
Effects of Mitocur-1 and Mitocur-3 on spontaneous and antigen-dependent degranulation. The cells were incubated with 1 µM of mitocurcuminoids for 3 h. To induce antigen-dependent degranulation, RBL-2H3 cells were sensitized by anti-DNP IgE and stimulated by DNP-BSA for 15 min. The level of degranulation was estimated by β-hexosaminidase release. The results are presented as the mean ± SD (*n* = 4). **** *p* ≤ 0.0001 as compared with the untreated DNP-BSA-stimulated cells calculated by one-way ANOVA, Dunnett’s test.

**Figure 3 ijms-24-01471-f003:**
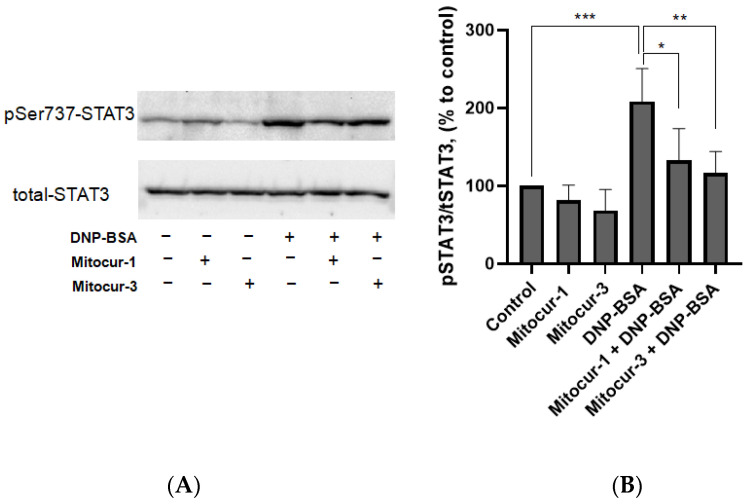
The effect of Mitocur-1 and Mitocur-3 on STAT3 serine 727 phosphorylation in RBL-2H3 cells. The cells were sensitized by anti-DNP IgE overnight, incubated in the presence of mitocurcuminoids for 3 h, stimulated with DNP-BSA for 20 min, and used for Western blot analysis. (**A**) Representative Western blots of cell lysates; (**B**) histograms denoting the relative amounts of proteins. The results are presented as the mean ± SD (*n* = 4). * *p* ≤ 0.05, ** *p* ≤ 0.01, *** *p* ≤ 0.001, one-way ANOVA, Tukey’s test.

**Figure 4 ijms-24-01471-f004:**
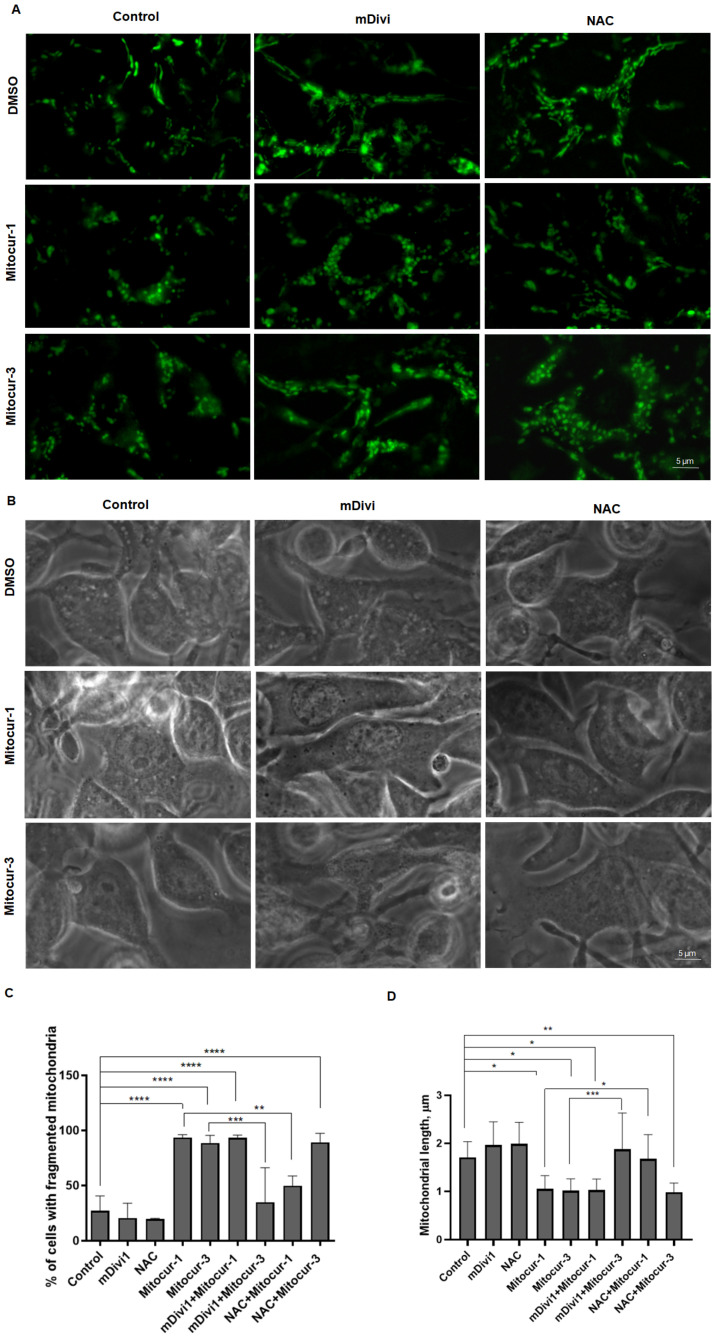
Effect of Mitocur-1 and Mitocur-3 on mitochondrial fragmentation. (**A**) Fluorescent microscopy of the mitochondria (labeled with MitoTracker Green) in the RBL-2H3 cells; (**B**) phase-contrast microscopy; (**C**) the percentage of cells with fragmented mitochondria. From 100 to 200 cells were analyzed in four independent experiments; (**D**) average mitochondrial length per cell. Twenty cells were analyzed in each group. The results are presented as the mean ± SD (*n* = 4). * *p* ≤ 0.05, ** *p* ≤ 0.01, *** *p* ≤ 0.001, and **** *p* ≤ 0.0001, one-way ANOVA, Tukey’s test.

**Figure 5 ijms-24-01471-f005:**
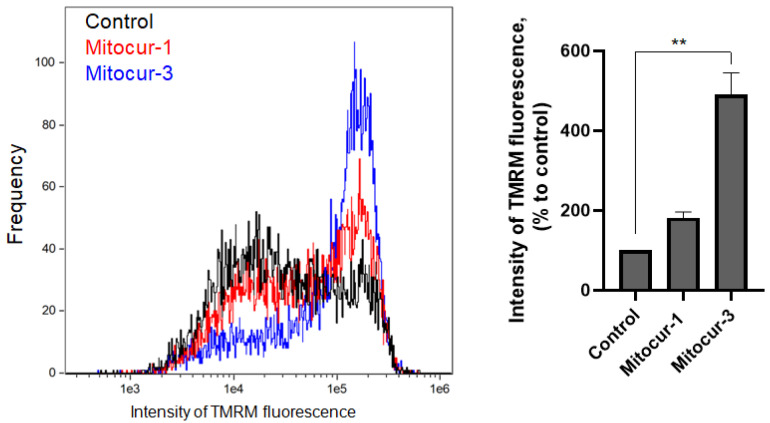
The effect of Mitocur-1 and Mitocur-3 on the mitochondrial membrane potential of RBL-2H3 cells. Cells were treated with compounds for 3 h. After treatment, cells were incubated with 50 nM TMRM for 30 min to detect mitochondrial membrane potential. The fluorescence in the PE channel was detected by flow cytometry. The results are presented as the mean ± SD (*n* = 4). ** *p* ≤ 0.01, as compared with the control cells calculated by one-way ANOVA, Dunnett’s test.

**Figure 6 ijms-24-01471-f006:**
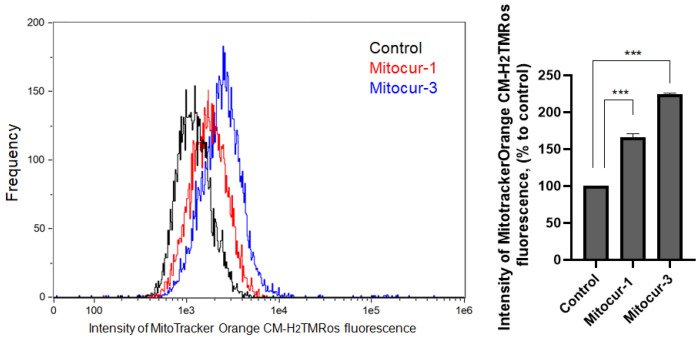
The effect of Mitocur-1 and Mitocur-3 on the ROS production of RBL-2H3 cells. Cells were treated with compounds for 3 h. After treatment, cells were incubated with 500 nM MitoTracker Orange CM-H2TMRos for 30 min and fluorescence in the PE channel was measured by flow cytometry. The results are presented as the mean ± SD (*n* = 4). *** *p* ≤ 0.001 as compared with the untreated non-stimulated cells calculated by one-way ANOVA, Dunnett’s test.

**Figure 7 ijms-24-01471-f007:**
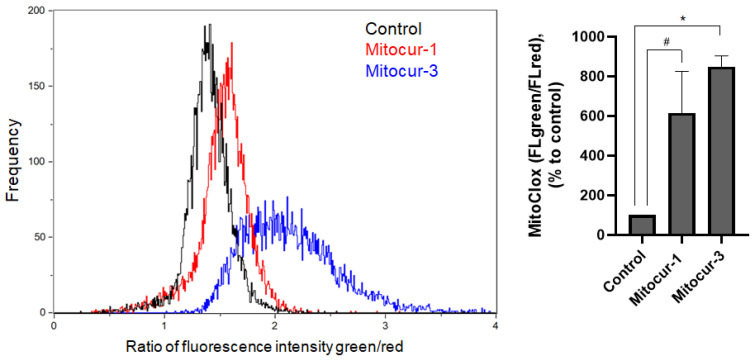
The effect of Mitocur-1 and Mitocur-3 on cardiolipin peroxidation in RBL-2H3 cells. Cells were incubated with the mitocurcuminoids for 3 h, washed with medium and 12 h later stained with MitoCLox for 30 min. MitoCLox fluorescence in the FITC and PE channels was measured by flow cytometry. The results are presented as the mean ± SD (*n* = 4). # *p* ≤ 0.1,* *p* ≤ 0.05 as compared with the control cells calculated by one-way ANOVA, Dunnett’s test.

**Figure 8 ijms-24-01471-f008:**
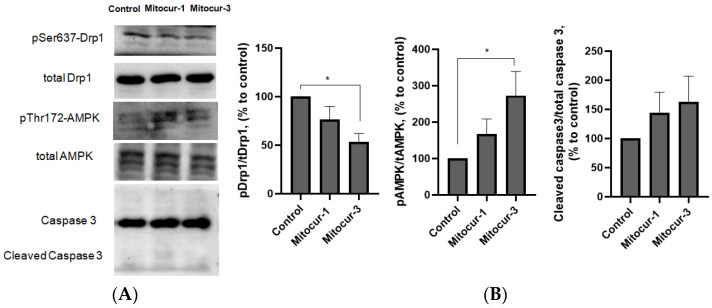
The effect of Mitocur-1 and Mitocur-3 on signaling molecules in RBL-2H3 cells. Cells were incubated in the presence of mitocurcuminoids for 3 h and used for Western blot analysis. (**A**) Representative Western blots of cell lysates; (**B**) histograms denoting the relative amounts of proteins. The results are presented as the mean ± SD (*n* ≥ 4). * *p* ≤ 0.05 when compared to untreated, non-stimulated cells, as determined by one-way ANOVA, Dunnett’s test.

**Figure 9 ijms-24-01471-f009:**
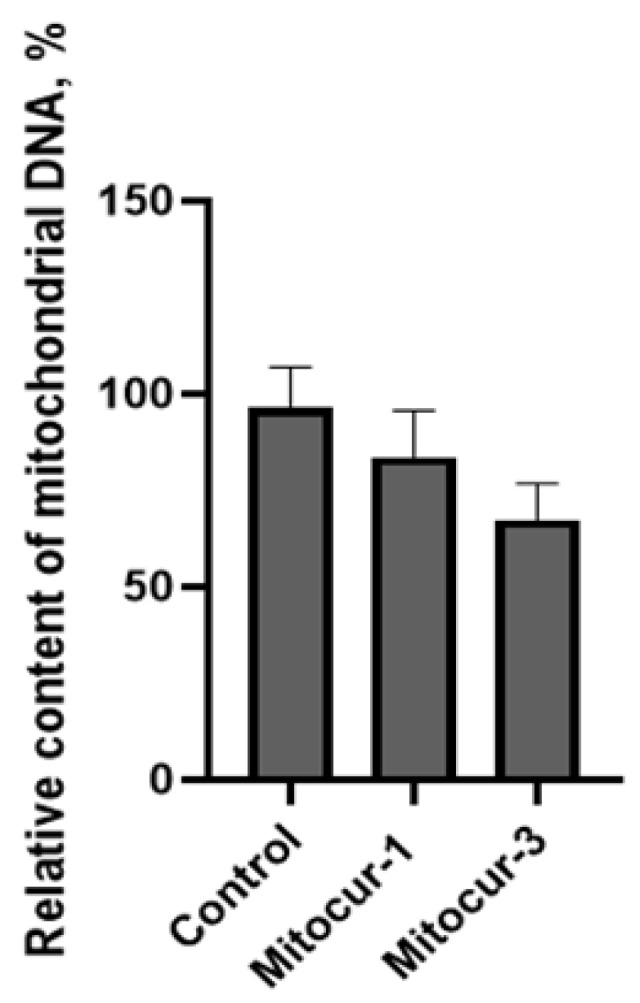
The effect of Mitocur-1 and Mitocur-3 on the relative mitochondrial DNA (mtDNA) content (ratio of mtDNA to nuclear DNA) in RBL-2H3 cells. Cells were incubated with mitocurcuminoids for 3 h, washed with medium and used for qPCR analysis 24 h later. The results are presented as the mean ± SD (*n* = 4).

## Data Availability

Not applicable.

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
