# Peer review of "The Distinct Effects of the Mitochondria-Targeted STAT3 Inhibitors Mitocur-1 and Mitocur-3 on Mast Cell and Mitochondrial Functions"

_ijms, 2023, doi:10.3390/ijms24021471_

Round 1

Reviewer 1 Report

The paper by Pavlyuchenkova and colleagues describes the results of their study where they investigated the effects of the mitochondria-targeted STAT3 inhibitors Mitocur-1 and Mitocur-3 on mast cell functions. This work follows a previous paper by Razin and colleagues, which demonstrated that these STAT3 inhibitors cause a significant reduction in mast cell exocytosis and cytokine release, due to a decrease in OXPHOS activity and STAT3 serine 727 phosphorylation.

In the present work, the authors point to differences in the mode of action of these two inhibitors. Specifically, they show that while both compounds cause mitochondrial fragmentation and both also increase mitochondrial ROS, inhibition of Drp1 prevented mitochondrial fragmentation induced by Mitocur-3 but not by Mitocur-1. They also show that the antioxidant N-acetylcysteine inhibits mitochondrial fission that is induced by Mitocur-1, but not by Mitocur-3. Finally, they show that mitochondrial fragmentation caused by Mitocur-3, but not by Mitocur-1, is accompanied by activation of Drp1 and AMPK as well as by an increase in the ΔΨm. Based on these data, the authors conclude that Mitocur-1 and Mitocur-3 have distinct mechanisms of action. Given the similarities in structure of these two inhibitors and the fact that both target mitochondrial STAT3, these unexpected results are novel and interesting. Following are a few comments that in my opinion will improve this manuscript:

1.     Figure 3: The authors should add experiments and statistical analysis to demonstrate that their results are significant.

2.     -Figure 4: To demonstrate that the inhibitors affect mitochondria dynamics, the authors should include images of the mitochondria taken at higher magnification. This is important as the authors base their conclusion on the incidence of cells that supposedly display fragmented mitochondria. However, it is not clear how they define a ‘fragmented mitochondrion’. I suggest that the authors quantify the average size of the mitochondria to show that the changes in mitochondria size are indeed significant.

-What happens in triggered cells in the absence and presence of the inhibitors?

-In the legend to this figure the authors refer to colours (green Mitotracker), but the image at least in my copy is black/white and mitochondria cannot really be identified.

Minor points:

1.     mkM in the legends to figures should be changed to μM

2.     Figure 3: the authors should specify in the legend for how long the cells were stimulated with DNP-BSA.

Author Response

Thank you very much for your suggestions, comments, and detailed evaluation.

  1. We've made additional Western blots and achieved statistical significance for the results presented on Fig.3.
  2. We've substituted the images with mitochondria at higher magnification (Fig.4A). We have also quantified the average size of the mitochondria and made a statistical analysis, confirming our previous observations (Fig. 4D).
  3. Concerning the triggered cells in the absence and presence of the inhibitors. The cells stimulated by DNP-BSA had highly fragmented mitochondria. Mitocur-1 and Mitocur-3 had no effect on this parameter.
  4. The images of mitochondria stained with MitoTracker Green on Fig.4 were taken with a B/W camera, thus they appeared without colours. We have made colour corrections by using ImageJ software. 

The minor points were corrected according to your suggestions.

Reviewer 2 Report

I find the manuscript suitable to be reconsidered for publication in IJMS after major revisions. Some of my suggestions are:

In the introduction, the authors limited themselves to considering only 14 works. I think the introduction needs to be expanded.

The pictures in figure 3 are very small, they should be reduced to the same size as figures 1 and 2.

The histogram in figure 4 is also very small.

Figures 5-9 need to unify the font for all figures.

Author Response

Thank you very much for your suggestions.

  1. We have significantly expanded the introduction.
  2. We have increased the size of Fig.3. 
  3. All the fonts for the figures were unified.

Round 2

Reviewer 2 Report

The authors made changes to the manuscript in accordance with my recommendations.

I believe that the article can be published in the present form.

Author Response

Thank you for the recommendations. We have performed additional editing of English language and style.